# Optimistic Gittins Indices

**Eli Gutin**
Operations Research Center, MIT
Cambridge, MA 02142
gutin@mit.edu

**Vivek F. Farias**
MIT Sloan School of Management
Cambridge, MA 02142
vivekf@mit.edu

## Abstract

Starting with the Thomspon sampling algorithm, recent years have seen a resurgence of interest in Bayesian algorithms for the Multi-armed Bandit (MAB) problem. These algorithms seek to exploit prior information on arm biases and while several have been shown to be regret optimal, their design has not emerged from a principled approach. In contrast, if one cared about Bayesian regret discounted over an infinite horizon at a *fixed, pre-specified* rate, the celebrated Gittins index theorem offers an *optimal* algorithm. Unfortunately, the Gittins analysis does not appear to carry over to minimizing Bayesian regret over all sufficiently large horizons and computing a Gittins index is onerous relative to essentially any incumbent index scheme for the Bayesian MAB problem.

The present paper proposes a sequence of 'optimistic' approximations to the Gittins index. We show that the use of these approximations in concert with the use of an increasing discount factor appears to offer a compelling alternative to state-of-the-art index schemes proposed for the Bayesian MAB problem in recent years by offering substantially improved performance with little to no additional computational overhead. In addition, we prove that the simplest of these approximations yields frequentist regret that matches the Lai-Robbins lower bound, including achieving matching constants.

## 1 Introduction

The multi-armed bandit (MAB) problem is perhaps the simplest example of a learning problem that exposes the tension between exploration and exploitation. Recent years have seen a resurgence of interest in Bayesian MAB problems wherein we are endowed with a prior on arm rewards, and a number of policies that exploit this prior have been proposed and/or analyzed. These include Thompson Sampling [20], Bayes-UCB [12], KL-UCB [9], and Information Directed Sampling [19]. The ultimate motivation for these algorithms appears to be two-fold: superior empirical performance and light computational burden. The strongest performance results available for these algorithms establish regret lower bounds that match the Lai-Robbins lower bound [15]. Even among this set of recently proposed algorithms, there is a wide spread in empirically observed performance.

Interestingly, the design of the index policies referenced above has been somewhat ad-hoc as opposed to having emerged from a principled analysis of the underlying Markov Decision process. Now if in contrast to requiring 'small' regret for all sufficiently large time horizons, we cared about minimizing Bayesian regret over an infinite horizon, discounted at a fixed, pre-specified rate (or equivalently, maximizing discounted infinite horizon rewards), the celebrated Gittins index theorem provides an *optimal, efficient* solution. Importing this celebrated result to the fundamental problem of designing algorithms that achieve low regret (either frequentist or Bayesian) simultaneously over all sufficiently large time horizons runs into two substantial challenges:

*High-Dimensional State Space:* Even minor 'tweaks' to the discounted infinite horizon objective render the corresponding Markov Decision problem for the Bayesian MAB problem intractable. For

instance, it is known that a Gittins-like index strategy is sub-optimal for a fixed horizon [5], let alone the problem of minimizing regret over all sufficiently large horizons.

*Computational Burden:* Even in the context of the discounted infinite horizon problem, the computational burden of calculating a Gittins index is substantially larger than that required for any of the index schemes for the multi-armed bandit discussed thus far.

The present paper attempts to make progress on these challenges. Specifically, we make the following contribution:

- We propose a class of 'optimistic' approximations to the Gittins index that can be computed with significantly less effort. In fact, the computation of the simplest of these approximations is no more burdensome than the computation of indices for the Bayes UCB algorithm, and several orders of magnitude faster than the nearest competitor, IDS.

- We establish that an arm selection rule that is greedy with respect to the simplest of these optimistic approximations achieves optimal regret in the sense of meeting the Lai-Robbins lower bound (including matching constants) provided the discount factor is increased at a certain rate.

- We show empirically that even the simplest optimistic approximation to the Gittins index proposed here *outperforms the state-of-the-art incumbent schemes discussed in this introduction by a non-trivial margin.* We view this as our primary contribution – the Bayesian MAB problem is fundamental making the performance improvements we demonstrate important.

**Literature review**    Thompson Sampling [20] was proposed as a heuristic to the MAB problem in 1933, but was largely ignored until the last decade. An empirical study by Chapelle and Li [7] highlighted Thompson Sampling's superior performance and led to a series of strong theoretical guarantees for the algorithm being proved in [2, 3, 12] (for specific cases when Gaussian and Beta priors are used). Recently, these proofs were generalized to the 1D exponential family of distributions in [13]. A few decades after Thompson Sampling was introduced, Gittins [10] showed that an index policy was optimal for the infinite horizon discounted MAB problem. Several different proofs for the optimality of Gittins index, were shown in [21, 22, 23, 6]. Inspired by this breakthrough, Lai and Robbins [15, 14], while ignoring the original MDP formulation, proved an asymptotic lower bound on achievable (non-discounted) regret and suggested policies that attained it.

Simple and efficient UCB algorithms were later developed by Agrawal and Auer et al. [1, 4], with finite time regret bounds. These were followed by the KL-UCB [9] and Bayes UCB [12] algorithms. The Bayes UCB paper drew attention to how well Bayesian algorithms performed in the frequentist setting. In that paper, the authors also demonstrated that a policy using indices similar to Gittins' had the lowest regret. The use of Bayesian techniques for bandits was explored further in [19] where the authors propose Information Directed Sampling, an algorithm that exploits complex information structures arising from the prior. There is also a very recent paper, [16], which also focuses on regret minimization using approximated Gittins Indices. However, in that paper, the time horizon is assumed to be known and fixed, which is different from the focus in this paper on finding a policy that has low regret over all sufficiently long horizons.

## 2    Model and Preliminaries

We consider a multi-armed bandit problem with a finite set of arms $\mathcal{A} = \{1, \ldots, A\}$. Arm $i \in \mathcal{A}$ if pulled at time $t$, generates a stochastic reward $X_{i,N_i(t)}$ where $N_i(t)$ denotes the cumulative number of pulls of arm $i$ up to and including time $t$. $(X_{i,s}, s \in \mathbb{N})$ is an i.i.d. sequence of random variables, each distributed according to $p_{\theta_i}(\cdot)$ where $\theta_i \in \Theta$ is a parameter. Denote by $\theta$ the tuple of all $\theta_i$. The expected reward from the $i^{\text{th}}$ arm is denoted by $\mu_i(\theta_i) := \mathbb{E}[X_{i,1} \mid \theta_i]$. We denote by $\mu^*(\theta)$ the maximum expected reward across arms; $\mu^*(\theta) := \max_i \mu_i(\theta_i)$ and let $i^*$ be an optimal arm. The present paper will focus on the Bayesian setting, and so we suppose that each $\theta_i$ is an independent draw from some prior distribution $q$ over $\Theta$. All random variables are defined on a common probability space $(\Omega, \mathcal{F}, \mathbb{P})$. We define a policy, $\pi := (\pi_t, t \in \mathbb{N})$, to be a stochastic process taking values in $\mathcal{A}$. We require that $\pi$ be adapted to the filtration $\mathcal{F}_t$ generated by the history of arm pulls and their corresponding rewards up to and including time $t - 1$.

Over time, the agent accumulates rewards, and we denote by

$$V(\pi, T, \theta) := \mathbb{E}\left[\sum_t X_{\pi_t, N_{\pi_t}(t)} \,\middle|\, \theta\right]$$

the reward accumulated up to time $T$ when using policy $\pi$. We write $V(\pi, T) := \mathbb{E}[V(\pi, T, \theta)]$. The regret of a policy over $T$ time periods, for a specific realization $\theta \in \Theta^A$, is the expected shortfall against always pulling the optimal arm, namely

$$\text{Regret}(\pi, T, \theta) := T\mu^*(\theta) - V(\pi, T, \theta)$$

In a seminal paper, [15], Lai and Robbins established a lower bound on achievable regret. They considered the class of policies under which for *any* choice of $\theta$ and positive constant $a$, any policy in the class achieves $o(n^a)$ regret. They showed that for any policy $\pi$ in this class, and any $\theta$ with a unique maximum, we must have

$$\liminf_T \frac{\text{Regret}(\pi, T, \theta)}{\log T} \geq \sum_i \frac{\mu^*(\theta) - \mu_i(\theta_i)}{d_{\text{KL}}(p_{\theta_i}, p_{\theta_{i*}})} \tag{1}$$

where $d_{\text{KL}}$ is the Kullback-Liebler divergence. The Bayes' risk (or Bayesian regret) is simply the expected regret over draws of $\theta$ according to the prior $q$:

$$\text{Regret}(\pi, T) := T\mathbb{E}[\mu^*(\theta)] - V(\pi, T).$$

In yet another landmark paper, [15] showed that for a restricted class of priors $q$ a similar class of algorithms to those found to be regret optimal in [14] were also Bayes optimal. Interestingly, however, this class of algorithms ignores information about the prior altogether. A number of algorithms that *do* exploit prior information have in recent years received a good deal of attention; these include Thompson sampling [20], Bayes-UCB [12], KL-UCB [9], and Information Directed Sampling [19].

The Bayesian setting endows us with the structure of a (high dimensional) Markov Decision process. An alternative objective to minimizing Bayes risk, is the maximization of the cumulative reward discounted over an infinite horizon. Specifically, for any positive discount factor $\gamma < 1$, define

$$V_\gamma(\pi) := \mathbb{E}_q\left[\sum_{t=1}^\infty \gamma^{t-1} X_{\pi_t, N_{\pi_t}(t)}\right].$$

The celebrated Gittins index theorem provides an *optimal, efficient* solution to this problem that we will describe in greater detail shortly; unfortunately as alluded to earlier even a minor 'tweak' to the objective above – such as maximizing cumulative expected reward over a finite horizon renders the Gittins index sub-optimal [17].

As a final point of notation, every scheme we consider will maintain a posterior on the mean of an arm at every point in time. We denote by $q_{i,s}$ the posterior on the mean of the $i$th arm after $s - 1$ pulls of that arm; $q_{i,1} := q$. Since our prior on $\theta_i$ will frequently be conjugate to the distribution of the reward $X_i$, $q_{i,s}$ will permit a succinct description via a sufficient statistic we will denote by $y_{i,s}$; denote the set of all such sufficient statistics $\mathcal{Y}$. We will thus use $q_{i,s}$ and $y_{i,s}$ interchangeably and refer to the latter as the 'state' of the $i$th arm after $s - 1$ pulls.

## 3 Gittins Indices and Optimistic Approximations

One way to compute the Gittins Index is via the so-called retirement value formulation [23]. The *Gittins Index* for arm $i$ in state $y$ is the value for $\lambda$ that solves

$$\frac{\lambda}{1 - \gamma} = \sup_{\tau > 1} \mathbb{E}\left[\sum_{t=1}^{\tau-1} \gamma^{t-1} X_{i,t} + \gamma^{\tau-1} \frac{\lambda}{1 - \gamma} \,\middle|\, y_{i,1} = y\right]. \tag{2}$$

We denote this quantity by $\nu_\gamma(y)$. If one thought of the notion of retiring as receiving a deterministic reward $\lambda$ in every period, then the value of $\lambda$ that solves the above equation could be interpreted as the per-period retirement reward that makes us indifferent between retiring immediately and the option of continuing to play arm $i$ with the potential of retiring at some future time. The Gittins index policy can thus succinctly be stated as follows: at time $t$, play an arm in the set $\arg\max_i \nu_\gamma(y_{i,N_i(t)})$. Ignoring computational considerations, we cannot hope for a scheme such as the one above to achieve acceptable regret or Bayes risk. Specifically, denoting the Gittins policy by $\pi^{G,\gamma}$, we have

**Lemma 3.1.** *There exists an instance of the multi armed bandit problem with $|\mathcal{A}| = 2$ for which*

$$\text{Regret}\left(\pi^{G,\gamma}, T\right) = \Omega(T)$$

*for any $\gamma \in (0, 1)$.*

The above result is expected. If the posterior means on the two arms are sufficiently apart, the Gittins index policy will pick the arm with the larger posterior mean. The threshold beyond which the Gittins policy 'exploits' depends on the discount factor and with a fixed discount factor there is a positive probability that the superior arm is never explored sufficiently so as to establish that it is, in fact, the superior arm. Fixing this issue then requires that the discount factor employed increase over time.

Consider then employing discount factors that increase at roughly the rate $1 - 1/t$; specifically, consider setting

$$\gamma_t = 1 - \frac{1}{2^{\lfloor \ln_2 t \rfloor + 1}}$$

and consider using the policy that at time $t$ picks an arm from the set $\arg\max_i \nu_{\gamma_t}(y_{i,N_i(t)})$. Denote this policy by $\pi^{\mathrm{D}}$. The following proposition shows that this 'doubling' policy achieves Bayes risk that is within a factor of $\log T$ of the optimal Bayes risk. Specifically, we have:

**Proposition 3.1.**

$$\text{Regret}(\pi^{\mathrm{D}}, T) = O\left(\log^3 T\right).$$

where the constant in the big-Oh term depends on the prior $q$ and $A$. The proof of this simple result (Appendix A.1) relies on showing that the finite horizon regret achieved by using a Gittins index with an appropriate fixed discount factor is within a constant factor of the optimal finite horizon regret. The second ingredient is a doubling trick.

While increasing discount factors does not appear to get us to the optimal Bayes risk (the achievable lower bound being $\log^2 T$; see [14]); we conjecture that in fact this is a deficiency in our analysis for Proposition 3.1. In any case, the policy $\pi^{\mathrm{D}}$ is not the primary subject of the paper but merely a motivation for the discount factor schedule proposed. Putting aside this issue, one is still left with the computational burden associated with $\pi^{\mathrm{D}}$ – which is clearly onerous relative to any of the incumbent index rules discussed in the introduction.

## 3.1 Optimistic Approximations to The Gittins Index

The retirement value formulation makes clear that computing a Gittins index is equivalent to solving a discounted, infinite horizon stopping problem. Since the state space $\mathcal{Y}$ associated with this problem is typically at least countable, solving this stopping problem, although not necessarily intractable, is a non-trivial computational task. Consider the following alternative stopping problem that requires as input the parameters $\lambda$ (which has the same interpretation as it did before), and $K$, an integer limiting the number of steps that we need to look ahead. For an arm in state $y$ (recall that the state specifies sufficient statistics for the current prior on the arm reward), let $R(y)$ be a random variable drawn from the prior on expected arm reward specified by $y$. Define the retirement value $R_{\lambda,K}(s, y)$ according to

$$R_{\lambda,K}(s, y) = \begin{cases} \lambda, & \text{if } s < K + 1 \\ \max\left(\lambda, R(y)\right), & \text{otherwise} \end{cases}$$

For a given $K$, the *Optimistic Gittins Index* for arm $i$ in state $y$ is now defined as the value for $\lambda$ that solves

$$\frac{\lambda}{1 - \gamma} = \sup_{1 < \tau \leq K+1} \mathbb{E}\left[\sum_{s=1}^{\tau-1} \gamma^{s-1} X_{i,s} + \gamma^{\tau-1} \frac{R_{\lambda,K}(\tau, y_{i,\tau})}{1 - \gamma} \,\middle|\, y_{i,1} = y\right]. \tag{3}$$

We denote the solution to this equation by $v_\gamma^K(y)$. The problem above admits a simple, attractive interpretation: nature reveals the *true* mean reward for the arm at time $K + 1$ should we choose to not retire prior to that time, which enables the decision maker to then instantaneously decide whether to retire at time $K + 1$ or else, never retire. In this manner one is better off than in the stopping problem inherent to the definition of the Gittins index, so that we use the moniker optimistic. Since we need to look ahead at most $K$ steps in solving the stopping problem implicit in the definition above, the computational burden in index computation is limited. The following Lemma formalizes this intuition

**Lemma 3.2.** *For all discount factors $\gamma$ and states $y \in \mathcal{Y}$, we have*

$$v_\gamma^K(y) \geq v_\gamma(y) \quad \forall K.$$

*Proof.* See Appendix A.2. □

It is instructive to consider the simplest version of the approximation proposed here, namely the case where $K = 1$. There, equation (3) simplifies to

$$\lambda = \hat{\mu}(y) + \gamma \mathbb{E}\left[(\lambda - R(y))^+\right] \tag{4}$$

where $\hat{\mu}(y) := \mathbb{E}\left[R(y)\right]$ is the mean reward under the prior given by $y$. The equation for $\lambda$ above can also be viewed as an upper confidence bound to an arm's expected reward. Solving equation (4) is often simple in practice, and we list a few examples to illustrate this:

**Example 3.1** (Beta). *In this case $y$ is the pair $(a, b)$, which specifies a Beta prior distribution. The 1-step Optimistic Gittins Index, is the value of $\lambda$ that solves*

$$\lambda = \frac{a}{a + b} + \gamma \mathbb{E}\left[(\lambda - \text{Beta}(a, b))^+\right] = \frac{a}{a + b}(1 - \gamma F_{a+1,b}^\beta(\lambda)) + \gamma \lambda(1 - F_{a,b}^\beta(\lambda))$$

*where $F_{a,b}^\beta$ is the CDF of a Beta distribution with parameters $a, b$.*

**Example 3.2** (Gaussian). *Here $y = (\mu, \sigma^2)$, which specifies a Gaussian prior and the corresponding equation is*

$$\lambda = \mu + \gamma \mathbb{E}\left[(\lambda - \mathcal{N}(\mu, \sigma^2))^+\right]$$
$$= \mu + \gamma\left[(\lambda - \mu)\Phi\left(\frac{\mu - \lambda}{\sigma}\right) + \sigma\phi\left(\frac{\mu - \lambda}{\sigma}\right)\right]$$

*where $\phi$ and $\Phi$ denote the Gaussian PDF and CDF, respectively.*

Notice that in both the Beta and Gaussian examples, the equations for $\lambda$ are in terms of distribution functions. Therefore it's straightforward to compute a derivative for these equations (which would be in terms of the density and CDF of the prior) which makes finding a solution, using a method such as Newton-Raphson, simple and efficient.

We summarize the Optimistic Gittins Index (OGI) algorithm succinctly as follows.

*Assume the state of arm $i$ at time $t$ is given by $y_{i,t}$, and let $\gamma_t = 1 - 1/t$. Play an arm*

$$i^* \in \arg\max_i v_{\gamma_t}^K(y_{i,t}),$$

*and update the posterior on the arm based on the observed reward.*

## 4 Analysis

We establish a regret bound for Optimistic Gittins Indices when the algorithm is given the parameter $K = 1$, the prior distribution $q$ is uniform and arm rewards are Bernoulli. The result shows that the algorithm, in that case, meets the Lai-Robbins lower bound and is thus asymptotically optimal, in both a frequentist and Bayesian sense. After stating the main theorem, we briefly discuss two generalizations to the algorithm.

In the sequel, whenever $x, y \in (0, 1)$, we will simplify notation and let $d(x, y) := d_{\text{KL}}(\text{Ber}(x), \text{Ber}(y))$. Also, we will refer to the Optimistic Gittins Index policy simply as $\pi^{\text{OG}}$, with the understanding that this refers to the case when $K$, the 'look-ahead' parameter, equals 1 and a flat beta prior is used. Moreover, we will denote the Optimistic Gittins Index of the $i^{\text{th}}$ arm as $v_{i,t} := v_{1-1/t}^1(y_{i,t})$. Now we state the main result:

**Theorem 1.** *Let $\epsilon > 0$. For the multi-armed bandit problem with Bernoulli rewards and any parameter vector $\theta \subset [0, 1]^A$, there exists $T^* = T^*(\epsilon, \theta)$ and $C = C(\epsilon, \theta)$ such that for all $T \geq T^*$,*

$$\text{Regret}\left(\pi^{\text{OG}}, T, \theta\right) \leq \sum_{\substack{i=1,\ldots,A \\ i \neq i^*}} \frac{(1 + \epsilon)^2(\theta^* - \theta_i)}{d(\theta_i, \theta^*)} \log T + C(\epsilon, \theta) \tag{5}$$

*where $C(\epsilon, \theta)$ is a constant that is only determined by $\epsilon$ and the parameter $\theta$.*

*Proof.* Because we prove frequentist regret, the first few steps of the proof will be similar to that of UCB and Thompson Sampling.

Assume w.l.o.g that arm 1 is uniquely optimal, and therefore $\theta^* = \theta_1$. Fix an arbitrary suboptimal arm, which for convenience we will say is arm 2. Let $j_t$ and $k_t$ denote the number of pulls of arms 1 and 2, respectively, by (but not including) time $t$. Finally, we let $s_t$ and $s'_t$ be the corresponding integer reward accumulated from arms 1 and 2, respectively. That is,

$$ s_t = \sum_{s=1}^{j_t} X_{1,s} \qquad s'_t = \sum_{s=1}^{k_t} X_{2,s}. $$

Therefore, by definition, $j_1 = k_1 = s_1 = s'_1 = 0$. Let $\eta_1, \eta_2, \eta_3 \in (\theta_2, \theta_1)$ be chosen such that $\eta_1 < \eta_2 < \eta_3$, $d(\eta_1, \eta_3) = \frac{d(\theta_2, \theta_1)}{1+\epsilon}$ and $d(\eta_2, \eta_3) = \frac{d(\eta_1, \eta_3)}{1+\epsilon}$. Next, we define $L(T) := \frac{\log T}{d(\eta_2, \eta_3)}$.

We upper bound the expected number of pulls of the second arm as follows,

$$ \mathbb{E}\left[k_T\right] \leq L(T) + \sum_{t=\lfloor L(T) \rfloor + 1}^{T} \mathbb{P}\left(\pi_t^{\mathrm{OG}} = 2,\ k_t \geq L(T)\right) $$

$$ \leq L(T) + \sum_{t=1}^{T} \mathbb{P}\left(v_{1,t} < \eta_3\right) + \sum_{t=1}^{T} \mathbb{P}\left(\pi_t^{\mathrm{OG}} = 2,\ v_{1,t} \geq \eta_3,\ k_t \geq L(T)\right) $$

$$ \leq L(T) + \sum_{t=1}^{T} \mathbb{P}\left(v_{1,t} < \eta_3\right) + \sum_{t=1}^{T} \mathbb{P}\left(\pi_t^{\mathrm{OG}} = 2,\ v_{2,t} \geq \eta_3,\ k_t \geq L(T)\right) $$

$$ \leq \frac{(1+\epsilon)^2 \log T}{d(\theta_2, \theta_1)} + \underbrace{\sum_{t=1}^{\infty} \mathbb{P}\left(v_{1,t} < \eta_3\right)}_{A} + \underbrace{\sum_{t=1}^{T} \mathbb{P}\left(\pi_t^{\mathrm{OG}} = 2,\ v_{2,t} \geq \eta_3,\ k_t \geq L(T)\right)}_{B} \quad (6) $$

All that remains is to show that terms $A$ and $B$ are bounded by constants. These bounds are given in Lemmas 4.1 and 4.2 whose proofs we describe at a high-level with the details in the Appendix.

**Lemma 4.1** (Bound on term A). *For any $\eta < \theta_1$, the following bound holds for some constant $C_1 = C_1(\epsilon, \theta_1)$*

$$ \sum_{t=1}^{\infty} \mathbb{P}\left(v_{1,t} < \eta\right) \leq C_1. $$

*Proof outline.* The goal is to bound $\mathbb{P}\left(v_{1,t} < \eta\right)$ by an expression that decays fast enough in $t$ so that the series converges. To prove this, we shall express the event $\{v_{1,t} < \eta\}$ in the form $\{W_t < 1/t\}$ for some sequence of random variables $W_t$. It turns out that for large enough $t$, $\mathbb{P}\left(W_t < 1/t\right) \leq \mathbb{P}\left(cU^{1/(1+h)} < 1/t\right)$ where $U$ is a uniform random variable, $c, h > 0$ and therefore $\mathbb{P}\left(v_{1,t} < \eta\right) = O\left(\frac{1}{t^{1+h}}\right)$. The full proof is in Appendix A.4.

We remark that the core technique in the proof of Lemma 4.1 is the use of the Beta CDF. As such, our analysis can, in some sense, improve the result for Bayes UCB. In the main theorem of [12], the authors state that the quantile in their algorithm is required to be $1 - 1/(t \log^c T)$ for some parameter $c \geq 5$, however they show simulations with the quantile $1 - 1/t$ and suggest that, in practice, it should be used instead. By utilizing techniques in our analysis, it is possible to prove that the use of $1 - 1/t$, as a discount factor, in Bayes UCB would lead to the same optimal regret bound. Therefore the 'scaling' by $\log^c T$ is unnecessary. □

**Lemma 4.2** (Bound on term B). *There exists $T^* = T^*(\epsilon, \theta)$ sufficiently large and a constant $C_2 = C_2(\epsilon, \theta_1, \theta_2)$ so that for any $T \geq T^*$, we have*

$$ \sum_{t=1}^{T} \mathbb{P}\left(\pi_t^{\mathrm{OG}} = 2,\ v_{2,t} \geq \eta_3,\ k_t \geq L(T)\right) \leq C_2. $$

*Proof outline.* This relies on a concentration of measure result and the assumption that the 2$^{\text{nd}}$ arm was sampled at least $L(T)$ times. The full proof is given in Appendix A.5. □

Lemma 4.1 and 4.2, together with (6), imply that

$$\mathbb{E}\left[k_T\right] \leq \frac{(1+\epsilon)^2 \log T}{d(\theta_2, \theta_1)} + C_1 + C_2$$

from which the regret bound follows. □

## 4.1 Generalizations and a tuning parameter

There is an argument in Agrawal and Goyal [2] which shows that any algorithm optimal for the Bernoulli bandit problem, can be modified to yield an algorithm that has $O(\log T)$ regret with general bounded stochastic rewards. Therefore Optimistic Gittins Indices is an effective and practical alternative to policies such as Thompson Sampling and UCB. We also suspect that the proof of Theorem 1 can be generalized to all lookahead values ($K > 1$) and to a general exponential family of distributions.

Another important observation is that the discount factor for Optimistic Gittins Indices does not have to be exactly $1 - 1/t$. In fact, a tuning parameter $\alpha > 0$ can be added to make the discount factor $\gamma_{t+\alpha} = 1 - 1/(t + \alpha)$ instead. An inspection of the proofs of Lemmas 4.1 and 4.2 shows that the result in Theorem 1 would still hold were one to use such a tuning parameter. In practice, performance is remarkably robust to our choice of $K$ and $\alpha$.

## 5 Experiments

Our goal is to benchmark Optimistic Gittins Indices (OGI) against state-of-the-art algorithms in the Bayesian setting. Specifically, we compare ourselves against Thomson sampling, Bayes UCB, and IDS. Each of these algorithms has in turn been shown to substantially dominate other extant schemes.

We consider the OGI algorithm for two values of the lookahead parameter $K$ (1 and 3) , and in one experiment included for completeness, the case of exact Gittins indices ($K = \infty$). We used a common discount factor schedule in all experiments setting $\gamma_t = 1 - 1/(100 + t)$. The choice of $\alpha = 100$ is second order and our conclusions remain unchanged (and actually appear to improve in an absolute sense) with other choices (we show this in a second set of experiments).

A major consideration in running the experiments is that the CPU time required to execute IDS (the closest competitor) based on the current suggested implementation is orders of magnitudes greater than that of the index schemes or Thompson Sampling. The main bottleneck is that IDS uses numerical integration, requiring the calculation of a CDF over, at least, hundreds of iterations. By contrast, the version of OGI with $K = 1$ uses 10 iterations of the Newton-Raphson method. In the remainder of this section, we discuss the results.

**Gaussian**   This experiment (Table 1) replicates one in [19]. Here the arms generate Gaussian rewards $X_{i,t} \sim \mathcal{N}(\theta_i, 1)$ where each $\theta_i$ is independently drawn from a standard Gaussian distribution. We simulate 1000 independent trials with 10 arms and 1000 time periods. The implementation of OGI in this experiment uses $K = 1$. It is difficult to compute exact Gittins indices in this setting, but a classical approximation for Gaussian bandits does exist; see [18], Chapter 6.1.3. We term the use of that approximation 'OGI($\infty$) Approx'. In addition to regret, we show the average CPU time taken, in seconds, to execute each trial.

| Algorithm | OGI(1) | OGI($\infty$) Approx. | IDS | TS | Bayes UCB |
|---|---|---|---|---|---|
| Mean Regret | 49.19 | 47.64 | 55.83 | 67.40 | 60.30 |
| S.D. | 51.07 | 50.59 | 65.88 | 47.38 | 45.35 |
| 1st quartile | 17.49 | 16.88 | 18.61 | 37.46 | 31.41 |
| Median | 41.72 | 40.99 | 40.79 | 63.06 | 57.71 |
| 3rd quartile | 73.24 | 72.26 | 78.76 | 94.52 | 86.40 |
| CPU time (s) | 0.02 | 0.01 | 11.18 | 0.01 | 0.02 |

Table 1: Gaussian experiment. OGI(1) denotes OGI with $K = 1$, while OGI Approx. uses the approximation to the Gaussian Gittins Index from [18].

The key feature of the results here is that OGI offers an approximately 10% improvement in regret over its nearest competitor IDS, and larger improvements (20 and 40 % respectively) over Bayes

UCB and Thompson Sampling. The best performing policy is OGI with the specialized Gaussian approximation since it gives a closer approximation to the Gittins Index. At the same time, OGI is essentially as fast as Thomspon sampling, and three orders of magnitude faster than its nearest competitor (in terms of regret).

**Bernoulli** In this experiment regret is simulated over 1000 periods, with 10 arms each having a uniformly distributed Bernoulli parameter, over 1000 independent trials (Table 2). We use the same setup as in [19] for consistency.

| Algorithm | OGI(1) | OGI(3) | OGI($\infty$) | IDS | TS | Bayes UCB |
|---|---|---|---|---|---|---|
| Mean Regret | 18.12 | 18.00 | 17.52 | 19.03 | 27.39 | 22.71 |
| 1st quartile | 6.26 | 5.60 | 4.45 | 5.85 | 14.62 | 10.09 |
| Median | 15.08 | 14.84 | 12.06 | 14.06 | 23.53 | 18.52 |
| 3rd quartile | 27.63 | 27.74 | 24.93 | 26.48 | 36.11 | 30.58 |
| CPU time (s) | 0.19 | 0.89 | (?) hours | 8.11 | 0.01 | 0.05 |

Table 2: Bernoulli experiment. OGI($K$) denotes the OGI algorithm with a $K$ step approximation and tuning parameter $\alpha = 100$. OGI($\infty$) is the algorithm that uses Gittins Indices.

Each version of OGI outperforms other algorithms and the one that uses (actual) Gittins Indices has the lowest mean regret. Perhaps, unsurprisingly, when OGI looks ahead 3 steps it performs marginally better than with a single step. Nevertheless, looking ahead 1 step is a reasonably close approximation to the Gittins Index in the Bernoulli problem. In fact the approximation error, when using an optimistic 1 step approximation, is around 15% and if $K$ is increased to 3, the error drops to around 4%.

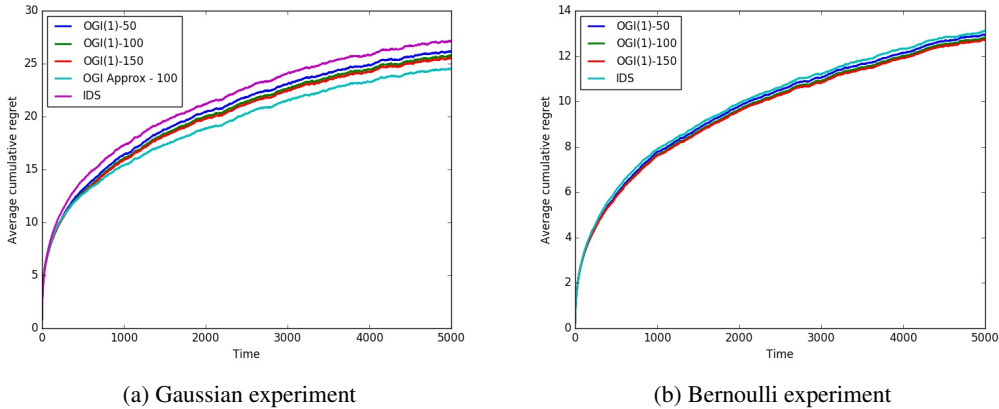

(a) Gaussian experiment          (b) Bernoulli experiment

Figure 1: Bayesian regret. In the legend, OGI($K$)-$\alpha$ is the format used to indicate parameters $K$ and $\alpha$. The OGI Appox policy uses the approximation to the Gittins index from [18].

**Longer Horizon and Robustness** For this experiment, we simulate the earlier Bernoulli and Gaussian bandit setups with a longer horizon of 5000 steps and with 3 arms. The arms' parameters are drawn at random in the same manner as the previous two experiments and regret is averaged over 100,000 independent trials. Results are shown in Figures 1a and 1b. In the Bernoulli experiment of this section, due to the computational cost, we are only able to simulate OGI with $K = 1$. In addition, to show robustness with respect to the choice of tuning parameter $\alpha$, we show results for $\alpha = 50, 100, 150$. The message here is essentially the same as in the earlier experiments: the OGI scheme offers a non-trivial performance improvement at a tiny fraction of the computational effort required by its nearest competitor. We omit Thompson Sampling and Bayes UCB from the plots in order to more clearly see the difference between OGI and IDS. The complete graphs can be found in Appendix A.6.

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
