[Supplementary Material · nips-2016-appendix.pdf]

# A  Appendix to Optimistic Gittins Indices

## A.1  Proof of Proposition 3.1

*Proof.* First, letting $\gamma_n = 1 - 1/n$, we show that

$$\text{Regret}\left(\pi^{G,\gamma_n}, n\right) = O\left(\log^2(n)\right). \tag{7}$$

Let $H \sim \text{Geo}(1/n)$ be an exogenous geometric random variable that is independent of $\theta$ and not observed by the agent. As an abbreviation, define $\mu^* = \mathbb{E}_q[\mu^*(\theta)]$. We then have

$$
\begin{aligned}
\sum_{t=1}^{\infty} \gamma^{t-1} \mathbb{E}\left[X_{\pi^{G,\gamma_n},t}\right] &= \mathbb{E}\left[\sum_{t=1}^{H} X_{\pi_t^{G,\gamma_n},t}\right] \\
&= \mathbb{E}\left[H\mu^*(\theta) - \text{Regret}\left(\pi^{G,\gamma_n}, H\right)\right] \\
&= n\mu^* - \mathbb{E}\left[\text{Regret}\left(\pi^{G,\gamma_n}, H\right)\right] \\
&\leq n\mu^* - \mathbb{E}\left[\text{Regret}\left(\pi^{G,\gamma_n}, H\right) \mid H > n\right] \mathbb{P}\left(H > n\right) \\
&\leq n\mu^* - \mathbb{E}\left[\text{Regret}\left(\pi^{G,\gamma_n}, n\right)\right] (1 - 1/n)^n \\
&= n\mu^* - \mathbb{E}\left[\text{Regret}\left(\pi^{G,\gamma_n}, n\right)\right] (e^{-1} + o(1)).
\end{aligned}
\tag{8}
$$
$$\tag{9}$$

Let $q, Q$ be the density and CDF, respectively, of the prior distribution. Now, by Theorem 3, part 1 of [14], there exists (an efficient) policy $\tilde{\pi}$, such that as $n$ becomes sufficiently large

$$\text{Regret}\left(\tilde{\pi}, n\right) \sim \left(A(A-1) \int_{\Theta} q^2(\theta) Q^{A-2}(\theta)\, d\theta\right) \log^2 n.$$

Therefore for some prior-dependent constant $C_q$, we have $\text{Regret}\left(\tilde{\pi}, n\right) \leq C_q \log^2 n$. Let $\Delta(\theta)$ denote worst case single period regret under parameter $\theta$, that is, $\Delta(\theta) = \max_i \mu(\theta^*) - \mu(\theta_i)$. Let $\Delta$ denote its expectation over $\theta$, from which we obtain the lower bound,

$$
\begin{aligned}
\mathbb{E}\left[\sum_{t=1}^{H} X_{\pi_t^{G,\gamma_n},t}\right] &\geq \mathbb{E}\left[\sum_{t=1}^{H} X_{\tilde{\pi}_t,t}\right] \tag{10}\\
&= \mathbb{E}\left[H\mu^*(\theta) - \text{Regret}\left(\tilde{\pi}, H\right)\right] \\
&\geq \mathbb{E}\left[H\mu^*(\theta) - \text{Regret}\left(\tilde{\pi}, H\right) \mathbb{1}\{H \geq e\} - 2\mathbb{1}\{H < e\}\Delta(\theta)\right] \\
&\geq n\mu^* - C_q \mathbb{E}\left[(\log(H))^2 \mathbb{1}\{H \geq 3\}\right] - 2\Delta \\
&\geq n\mu^* - C_q \mathbb{E}\left[(\log(H))^2 \mid H \geq 3\right] \mathbb{P}\left(H \geq 3\right) - 2\Delta \\
&\geq n\mu^* - C_q \log^2(n+3)\mathbb{P}\left(H \geq 3\right) - 2\Delta \tag{11}
\end{aligned}
$$

where (10) holds by optimality of the Gittins Index. The bound (11) follows from the memoryless property of the Geometric distribution, from Jensen's inequality and the fact that function $\log^2 x$ is a concave function on $[e, +\infty)$. Thus, equation (7) is implied by the bounds (9) and (11).

Now, for any policy $\pi$, we define $\tilde{L}_{\pi}(m) := m\mu^* - \sum_{t=1}^{m} X_{\pi_t,t}$ to be the random $m$ period shortfall against the expected Bayes' optimal arm and let $g_k = 1 - 1/2^{k-1}$. We break up the time horizon $T$

into geometrically growing epochs and bound, conservatively, the Bayes' risk in each one:

$$\text{Regret}\left(\pi^D, T\right) \leq \text{Regret}\left(\pi^D, 2^{\lceil \log_2 T \rceil}\right) \tag{12}$$

$$= \sum_{k=1}^{\lceil \log_2 T \rceil} \mathbb{E}\left[\mathbb{E}\left[\tilde{L}_{\pi^D}(2^{k-1}) \,\middle|\, \mathcal{F}_{2^{k-1}-1}\right]\right]$$

$$= \sum_{k=1}^{\lceil \log_2 T \rceil} \mathbb{E}\left[\mathbb{E}\left[\tilde{L}_{\pi^{G,g_k}}(2^{k-1}) \,\middle|\, \mathcal{F}_{2^{k-1}-1}\right]\right]$$

$$\leq \sum_{k=1}^{\lceil \log_2 T \rceil} \mathbb{E}\left[\mathbb{E}\left[\tilde{L}_{\pi^{G,g_k}}(2^{k-1}) \,\middle|\, \mathcal{F}_0\right]\right] \tag{13}$$

$$= \sum_{k=1}^{\lceil \log_2 T \rceil} \text{Regret}\left(\pi^{G,g_k}, 2^{k-1}\right) = O\left(\sum_{k=1}^{\lceil \log_2 T \rceil} k^2\right) \tag{14}$$

$$= O(\log^3 T)$$

where (14) follows from equation (7) and (13) holds because regret increases if history is discarded. $\square$

## A.2   Proof of Lemma 3.2

*Proof.* Fix $\lambda > 0$ and an arm $i$. Let $V_\lambda(y)$ be the value of the RHS of (2) with the per-period reward of $\lambda$, and define $\hat{V}_\lambda^K(y)$, similarly, for problem (3) (where $y$ is, as before, the state of an arm). Notice that because rewards are generated according to an unknown parameter $\theta_i$, which needs to be learned, that if we condition on a fixed $\theta_i$, we have for any stopping time $\tau$ that

$$\mathbb{E}\left[\sum_{t=1}^{\tau-1} \gamma^{t-1} X_{i,t} + \gamma^{\tau-1}\frac{\lambda}{1-\gamma} \,\middle|\, \theta_i\right] \leq \mathbb{E}\left[\sum_{t=1}^{\tau-1} \gamma^{t-1} \mu(\theta_i) + \gamma^{\tau-1}\frac{\lambda}{1-\gamma} \,\middle|\, \theta_i\right] \tag{15}$$

where the expectation is also taken over the agent's prior on $\theta_i$. Simply put, the best performance in the bandit game can be achieved if the parameter governing expected rewards is known from the beginning by the agent. Now recall that $R(y_{i,t})$ is a random variable drawn from the prior on the arm's mean reward at time $t$. We also define the function

$$r_{\lambda,K}(t,x) = \begin{cases} \lambda & t < K \\ \max(x,\lambda) & \text{otherwise} \end{cases}$$

Let $\tau$ be the stopping time at which the agent retires and define $\tau_K = \tau \wedge (K+1)$. We then bound $V_\lambda(y)$,

$$
\begin{aligned}
V_\lambda(y) &= \sup_{\tau>1} \mathbb{E}\left[\sum_{t=1}^{\tau-1} \gamma^{t-1}X_{i,t} + \gamma^{\tau-1}\frac{\lambda}{1-\gamma} \,\middle|\, y_{i,1}=y\right] \\
&= \sup_{\tau>1} \mathbb{E}\left[\mathbb{E}\left[\sum_{t=1}^{\tau-1} \gamma^{t-1}X_{i,t} + \gamma^{\tau-1}\frac{\lambda}{1-\gamma} \,\middle|\, \theta_i\right] \,\middle|\, y_{i,1}=y\right] \\
&= \sup_{\tau>1} \mathbb{E}\left[\sum_{t=1}^{\tau_K-1} \gamma^{t-1}X_{i,t} + \mathbb{E}\left[\sum_{t=\tau_K}^{\tau-1} \gamma^{t-1}X_{i,t} + \gamma^{\tau-1}\frac{\lambda}{1-\gamma} \,\middle|\, \theta_i\right] \,\middle|\, y_{i,1}=y\right] \\
&\leq \sup_{\tau>1} \mathbb{E}\left[\sum_{t=1}^{\tau_K-1} \gamma^{t-1}X_{i,t} + \mathbb{E}\left[\sum_{t=\tau_K}^{\tau-1} \gamma^{t-1}\mu(\theta_i) + \gamma^{\tau-1}\frac{\lambda}{1-\gamma} \,\middle|\, \theta_i\right] \,\middle|\, y_{i,1}=y\right] \quad (16) \\
&= \sup_{\tau>1} \mathbb{E}\left[\sum_{t=1}^{\tau_K-1} \gamma^{t-1}X_{i,t} + \mathbb{E}\left[\gamma^{\tau_K-1}\frac{r_{\lambda,K}(\tau_K, \mu(\theta_i))}{1-\gamma} \,\middle|\, \theta_i\right] \,\middle|\, y_{i,1}=y\right] \\
&= \sup_{\tau>1} \mathbb{E}\left[\sum_{t=1}^{\tau_K-1} \gamma^{t-1}X_{i,t} + \frac{\gamma^{\tau_K-1}}{1-\gamma}\mathbb{E}\left[\mathbb{E}\left[r_{\lambda,K}(\tau_K, \mu(\theta_i))\mid\theta_i\right]\mid\mathcal{F}_{\tau_K-1}\right]\right] \\
&= \sup_{\tau>1} \mathbb{E}\left[\sum_{t=1}^{\tau_K-1} \gamma^{t-1}X_{i,t} + \frac{\gamma^{\tau_K-1}}{1-\gamma}\underbrace{\mathbb{E}\left[r_{\lambda,K}(\tau_K, R(y_{i,\tau_K}))\right]}_{=R_{\lambda,K}(\tau_K, y_{i,\tau_K})} \,\middle|\, y_{i,1}=y\right] \\
&= \sup_{1<\tau\leq K+1} \mathbb{E}\left[\sum_{t=1}^{\tau} \gamma^{t-1}X_{i,t} + \gamma^{\tau-1}\frac{R_{\lambda,K}(\tau, y_{i,\tau})}{1-\gamma} \,\middle|\, y_{i,1}=y\right] = \hat{V}_\lambda^K(y).
\end{aligned}
$$

The main step in the above is (16) where we bound on the inner conditional expection (in terms of $\theta_i$) by applying (15). We also used the fact that $\mu(\theta_i) \mid \mathcal{F}_{t-1} \sim R(y_{i,t})$ for all $t$. Finally observe that both $\hat{V}_\lambda^K(y)$ and $V_\lambda(y)$ are increasing in $\lambda$ for any fixed $y$. Therefore if $\lambda_1 = (1-\gamma)\hat{V}_{\lambda_1}^K(y)$ and $\lambda_2 = (1-\gamma)V_{\lambda_2}(y)$, then, because $V_\lambda(y) \leq \hat{V}_\lambda^K(y)$ for any $\lambda$, it must be that $\lambda_1 \geq \lambda_2$. A simple argument shows this, which we omit. $\qquad\square$

### A.3   Results for the frequentist regret bound proof

#### A.3.1   Definitions and properties of Binomial distributions

We list notation and facts related to Beta and Binomial distributions, which are used through this section.

**Definition A.1.** $F_{n,p}^B(.)$ *is the CDF of the Binomial distribution with parameters $n$ and $p$, and $F_{a,b}^\beta(.)$ is the CDF of the Beta distribution with parameters $a$ and $b$.*

**Fact A.1.** *Let $a$ and $b$ be positive integers and $y \in [0,1]$,*

$$F_{a,b}^\beta(y) = 1 - F_{a+b-1,y}^B(a-1)$$

*Proof.* Proof is found in [2]. $\qquad\square$

**Fact A.2.** *The median of a Binomial$(n,p)$ distribution is either $\lceil np \rceil$ or $\lfloor np \rfloor$.*

*Proof.* Proof is found in [11]. $\qquad\square$

**Corollary A.1** (Corollary of Fact A.2). *Let $n$ be a positive integer and $p \in (0,1)$. For any nonnegative integer $s < np$*

$$F_{n,p}(s) \leq 1/2$$

**Fact A.3.** *Let $n$ be a positive integer and $p \in [0,1]$. Then for any $k \in \{0,\dots,n\}$,*

$$(1-p)F_{n-1,p}(k) \leq F_{n,p}(k) \leq F_{n-1,p}^B(k)$$

*Proof.* To prove $F_{n,p}(k) \leq F_{n-1,p}^B(k)$, we let $X_1, \ldots, X_n$ be i.i.d samples from a Bernoulli($p$) distribution. We then have

$$F_{n,p}^B(k) = \mathbb{P}\left(\sum_{i=1}^n X_i \leq k\right) \leq \mathbb{P}\left(\sum_{i=1}^{n-1} X_i \leq k\right) = F_{n-1,p}^B(k)$$

Now to prove $(1-p)F_{n-1,p}(k) \leq F_{n,p}(k)$, it's enough to observe that $F_{n,p}(k) = pF_{n-1,p}(k-1) + (1-p)F_{n-1,p}(k)$. $\qquad\square$

### A.3.2 Ratio of Binomial CDFs

**Lemma A.1.** *Let $0 < q < p < 1$. Let $n$ be a positive integer such that $e^{\frac{n}{2}d(q,p)} \geq (n+1)^4$ and let $k$ be a nonnegative integer such that $k < nq$. It then follows that*

$$F_{n,q}^B(k)/F_{n,p}^B(k) > e^{\frac{n}{2}d(q,p)}.$$

*Proof.* From the method of types (see [8]), we have for any $r \in (0,1)$ and $j < nr$

$$\frac{e^{-nd(j/n,r)}}{(1+n)^2} \leq F_{n,r}(j) \leq (n+1)^2 e^{-nd(j/n,r)}. \tag{17}$$

Because $k < nq < np$, by applying (17) to both the numerator and denominator, we get

$$\frac{F_{n,q}(k)}{F_{n,p}(k)} \geq \frac{e^{-nd(k/n,q)}}{(n+1)^4 e^{-nd(k/n,p)}} = \frac{e^{n(d(k/n,p)-d(k/n,q))}}{(n+1)^4}.$$

Examining the exponent, we find

$$\begin{aligned} d(k/n, p) - d(k/n, q) &= \frac{k}{n}\log\frac{q}{p} + \left(1 - \frac{k}{n}\right)\log\frac{1-q}{1-p} \\ &> q\log\frac{q}{p} + (1-q)\log\frac{1-q}{1-p} \\ &= d(q,p) \end{aligned}$$

where the bound holds because the expression is decreasing in $k$, and $k < nq$. Therefore,

$$\frac{F_{n,q}(k)}{F_{n,p}(k)} > \frac{e^{nd(q,p)}}{(n+1)^4} = \frac{e^{\frac{n}{2}d(q,p)}}{(n+1)^4} e^{\frac{n}{2}d(q,p)} \geq e^{\frac{n}{2}d(q,p)}. \tag{18}$$

The final lower bound in (18) follows from the assumption on $n$. $\qquad\square$

### A.3.3 Optimistic Gittins Index results

**Lemma A.2.** *Let $\gamma \in (0,1)$ and*

$$\lambda = \sup\{x \in [0,1] : \mathbb{E}[V] + \gamma\mathbb{E}\left[(x-V)^+\right] \geq x\} \tag{19}$$

*where $V$ is a continuous random variable with support $[0,1]$ and $\mathbb{E}[V] > 0$. For all $y \in (0,1)$, the following equivalence holds*

$$\lambda < y \iff \mathbb{E}[V] + \gamma\mathbb{E}\left[(y-V)^+\right] < y. \tag{20}$$

*Proof.* As a shorthand let $g(z) = \mathbb{E}[V] + \gamma\mathbb{E}[(z-V)^+]$. First let's assume $\lambda < y$. If $y \leq g(y)$, then $\lambda$ would not be the supremum over all real numbers $z \in [0,1]$ such that $z \leq g(z)$. Therefore $g(y) < y$.

For the converse, assume $\lambda \geq y$. Observe that $g(z)$ is convex. Also, one can verify through the bounded convergence theorem that $g(z)$ is differentiable [the event $\{V = z\}$, at which $(z-V)^+$ is not differentiable, has measure zero]. Thirdly, because $g(.)$ is continuous on $[0,1]$, by the Intermediate Value Theorem, it has a fixed point and in particular $\lambda = g(\lambda)$. Therefore let $\epsilon < (1-\lambda)/2$ and from the first direction of the proof, we have $\lambda + \epsilon > g(\lambda + \epsilon)$. Thus

$$g(\lambda + \epsilon) \geq g(\lambda) + \epsilon g'(\lambda) = \lambda + \epsilon g'(\lambda)$$

where the inequality follows from $g(.)$ being convex and differentiable. This implies that $g'(\lambda) < 1$ and, moreover, because $g(z)$ is also increasing, it follows that $g'(\lambda) \in (0, 1)$, whence

$$\begin{aligned}
g(y) &\geq g(\lambda) - (\lambda - y)g'(\lambda) \\
&= \lambda - (\lambda - y)g'(\lambda) \\
&= (1 - g'(\lambda))\lambda + g'(\lambda)y \\
&\geq \min(y, \lambda) = y.
\end{aligned}$$

$\square$

**Corollary A.2.** *Let $v_{i,t}$ be the approximate Optimistic Gittins Index, under the Bernoulli problem with uniform priors, of arm $i$ at time $t$ and let $x \in (0, 1)$. The following equivalence holds*

$$\{v_{i,t} < x\} = \{\mathbb{E}[V_t] + \gamma_t \mathbb{E}[(x - V_t)^+] < x\}$$

*where $V_t \sim Beta(s_t + 1, j_t - s_t + 1)$, $j_t$ denotes the number of pulls of arm $i$ and $s_t$ the number of successes observed.*

*Proof.* By the definition in Equation (4), $v_{i,t}$ can be characterized with the relation

$$v_{i,t} = \sup\{u \in [0, 1] : u \leq \mathbb{E}[V_t] + \gamma_t \mathbb{E}[(u - V_t)^+]\}.$$

The conclusion then follows from Lemma A.2. $\square$

### A.4   Proof of Lemma 4.1

*Proof.* Define $\delta := (\theta_1 - \eta)/2$ and $\eta' := \eta + \delta$. In other words, $\delta$ is half the distance between $\eta$ and $\theta_1$; $\eta'$ is the point half-way. The proof consists of showing two claims

**Claim 1:** $\{v_{1,t} < \eta\} \subseteq \{F^B_{j_t+1,\eta'}(s_t) < \frac{1}{\delta t}\}$**:**

Let $V_t \sim Beta(s_t + 1, j_t - s_t + 1)$ be the agent's posterior on the optimal arm. Using Corollary A.2, we find that

$$\begin{aligned}
\{v_{1,t} < \eta\} &= \{\mathbb{E}[V_t] + \gamma_t \mathbb{E}[(\eta - V_t)^+] < y\} \\
&= \left\{\mathbb{E}[(V_t - \eta)^+] < \frac{1}{t}\right\}
\end{aligned} \tag{21}$$

where the second equality is obtained from rearranging terms. We approximate the conditional expectation in (21) with

$$\begin{aligned}
\mathbb{E}[(V_t - \eta)^+ \mid s_t, j_t] &= \mathbb{E}[(V_t - \eta)\mathbb{1}\{V_t \geq \eta\}] \\
&= \mathbb{E}[(V_t - \eta)\mathbb{1}\{\eta + \delta > V_t \geq \eta\}] \\
&\quad + \mathbb{E}[(V_t - \eta)\mathbb{1}\{V_t \geq \eta + \delta\}] \\
&> \mathbb{E}[(V_t - \eta)\mathbb{1}\{V_t \geq \eta + \delta\}] \\
&\geq \delta\mathbb{P}(V_t \geq \eta') \\
&= \delta(1 - F_{s_t+1,j_t-s_t+1}(\eta')) = \delta F^B_{j_t+1,\eta'}(s_t)
\end{aligned} \tag{22}$$

The last equality is due to Fact A.1 and this proves the claim.

**Claim 2:** $\sum_{t=1}^{\infty} \mathbb{P}\left(F^B_{j_t+1,\eta'}(s_t) < \frac{1}{\delta t}\right) \leq C_1$ **where $C_1$ is a constant:**

Let us fix the sequence $f_t = -\frac{\log \delta t}{\log(1-\eta')} - 1 = O(\log t)$. We then have

$$\begin{aligned}
\mathbb{P}\left(F^B_{j_t+1,\eta'}(s_t) < \frac{1}{\delta t}\right) &= \mathbb{P}\left(F^B_{j_t+1,\eta'}(s_t) < \frac{1}{\delta t}, \ j_t > f_t\right) \\
&\quad + \mathbb{P}\left(F^B_{j_t+1,\eta'}(s_t) < \frac{1}{\delta t}, \ j_t \leq f_t\right).
\end{aligned} \tag{23}$$

For the second term in the RHS of (23) we have the following bound,

$$\mathbb{P}\left(F^B_{j_t+1,\eta'}(s_t) < \frac{1}{\delta t}, \; j_t \le f_t\right) \le \mathbb{P}\left(F^B_{j_t+1,\eta'}(0) < \frac{1}{\delta t}, \; j_t \le f_t\right)$$

$$= \mathbb{P}\left((1-\eta')^{j_t+1} < \frac{1}{\delta t}, \; j_t \le f_t\right)$$

$$\le \mathbb{P}\left((1-\eta')^{f_t+1} < \frac{1}{\delta t}\right) = 0. \tag{24}$$

Now we use the following fact to bound the left term on the RHS of (23). Define the function

$$F^{-B}_{n,p}(u) := \inf\{x : F^B_{n,p}(x) \ge u\}$$

which is the inverse CDF. Then it is known that if $U \sim \text{Unif}(0,1)$, then $F^{-B}_{n,p}(U) \sim \text{Binomial}(n,p)$. Furthermore, $F^B_{n,p}(F^{-B}_{n,p}(U)) \ge U$ due to the definition of the inverse CDF.

Now let us only consider large $t$, in particular $t > M = M(\theta_1, \eta')$ where:

1. $M$ is such that $e^{d(\eta',\theta_1)f_M/2} > (f_M + 1)^4$

2. $M > \frac{4}{(1-\eta')\delta}$

3. $\lceil f_M \rceil > 0$ and $F^B_{\lceil f_M \rceil,\eta'}(f_M \eta') > 1/4$. Note that there is a large enough integer for this because $F^B_{\lceil f_t \rceil,\eta'}(f_t \eta') \to \frac{1}{2}$ as $t \to \infty$.

Suppose that $t > M$. It then follows that the event $\{F^B_{j_t,\eta'}(s_t) < \frac{1}{(1-\eta')\delta t}, \; s_t \ge j_t\eta', \; j_t > f_t\}$ has measure zero because of the assumptions made on $M$. Therefore if $t > M$, we have

$$\mathbb{P}\left(F^B_{j_t+1,\eta'}(s_t) < \frac{1}{\delta t}, \; j_t > f_t\right)$$

$$\le \mathbb{P}\left(F^B_{j_t,\eta'}(s_t) < \frac{1}{(1-\eta')\delta t}, \; j_t > f_t\right) \tag{25}$$

$$= \mathbb{P}\left(F^B_{j_t,\eta'}(s_t) < \frac{1}{(1-\eta')\delta t}, \; s_t < j_t\eta', \; j_t > f_t\right)$$

$$= \mathbb{P}\left(F^B_{j_t,\theta_1}(s_t)\frac{F^B_{j_t,\eta'}(s_t)}{F^B_{j_t,\theta_1}(s_t)} < \frac{1}{(1-\eta')\delta t}, \; s_t < j_t\eta', \; j_t > f_t\right)$$

$$\le \mathbb{P}\left(F^B_{j_t,\theta_1}(s_t)e^{j_t D} < \frac{1}{(1-\eta')\delta t}, \; j_t > f_t\right) \tag{26}$$

$$\le \mathbb{P}\left(F^B_{j_t,\theta_1}(s_t)e^{f_t D} < \frac{1}{(1-\eta')\delta t}\right)$$

$$= \mathbb{P}\left(F^B_{j_t,\theta_1}(F^{-B}_{j_t,\theta_1}(U)) < \frac{e^{-f_t D}}{(1-\eta')\delta t}\right) \tag{27}$$

$$\le \mathbb{P}\left(U < \frac{e^{-f_t D}}{(1-\eta')\delta t}\right)$$

$$= \frac{e^{-f_t D}}{(1-\eta')\delta t}$$

$$= O\left(\frac{1}{t^{1+Dc_{\eta'}}}\right) \tag{28}$$

where $D = d(\eta', \theta_1) > 0$ and $c_{\eta'} = -\log^{-1}(1-\eta') > 0$ are constant. The bound (25) holds due to Fact (A.3). Bound (26) follows from an application of Lemma A.1 and the fact that $t > M$. Equation (27) follows from $s_t \sim \text{Binomial}(j_t, \theta_1)$ and the inverse sampling technique. By combining bounds

(28), (24) and (23), we get

$$\sum_{t=1}^{\infty} \mathbb{P}\left(F_{j_t+1,\eta'}^B(s_t) < \frac{1}{\delta t}\right) \leq M + \sum_{t=M+1}^{\infty} \mathbb{P}\left(F_{j_t+1,\eta'}^B(s_t) < \frac{1}{\delta t}\right) \leq M + C_1' =: C_1$$

where $C_1' = C_1'(\theta_1, \eta')$ is some other constant, namely the limit of the series. $\qquad\square$

## A.5 Proof of Lemma 4.2

*Proof.* See the main proof of Theorem 1 to recall the definition of constants $\eta_1$, $\eta_3$ and their relationship with $\theta_2$ and $\theta_1$. As an abbreviation we let $L = L(T)$.

Firstly, by the law of total probability, we find that

$$\sum_{t=1}^{T} \mathbb{P}(v_{2,t} \geq \eta_3, \ k_t \geq L, \ \pi_t^{\text{OG}} = 2)$$

$$= \sum_{t=1}^{T} \mathbb{P}\left(v_{2,t} \geq \eta_3, \ k_t \geq L, \ s_t' < \lfloor k_t\eta_1 \rfloor, \ \pi_t^{\text{OG}} = 2\right)$$

$$+ \sum_{t=1}^{T} \mathbb{P}\left(v_{2,t} \geq \eta_3, \ k_t \geq L, \ s_t' \geq \lfloor k_t\eta_1 \rfloor, \ \pi_t^{\text{OG}} = 2\right)$$

$$\leq \sum_{t=1}^{T} \mathbb{P}\left(v_{2,t} \geq \eta_3, \ k_t \geq L, \ s_t' < \lfloor k_t\eta_1 \rfloor\right) + \sum_{t=1}^{T} \mathbb{P}\left(\pi_t^{\text{OG}} = 2, \ s_t' \geq \lfloor k_t\eta_1 \rfloor\right) \quad (29)$$

Let $V_t \sim \text{Beta}(s_t' + 1, k_t - s_t' + 1)$ denote the agent's posterior on the second arm at time $t$, then

$$\sum_{t=1}^{T} \mathbb{P}(v_{2,t} \geq \eta_3, \ k_t \geq L, \ s_t' < \lfloor k_t\eta_1 \rfloor)$$

$$= \sum_{t=1}^{T} \mathbb{P}\left(\mathbb{E}[V_t] + \gamma_t \mathbb{E}\left[(\eta_3 - V_t)^+\right] \geq \eta_3, \ k_t \geq L, \ s_t' < \lfloor k_t\eta_1 \rfloor\right)$$

$$= \sum_{t=1}^{T} \mathbb{P}\left(\frac{\mathbb{E}\left[(\eta_3 - V_t)^+\right]}{\mathbb{E}\left[(V_t - \eta_3)^+\right]} \leq t, \ k_t \geq L, \ s_t' < \lfloor k_t\eta_1 \rfloor\right) \quad (30)$$

where the second equality follows from Corollary A.2 in Appendix A.3.3. The following result lets us bound (30),

**Lemma A.3.** *Let $0 < x < y < 1$. For any nonnegative integers $s$ and $k$ with $s < \lfloor kx \rfloor$, it holds that*

$$\frac{\mathbb{E}\left[(y - V)^+\right]}{\mathbb{E}\left[(V - y)^+\right]} \geq \frac{(y - x)\exp(kd(x,y))}{2}$$

*where $V \sim \text{Beta}(s + 1, k - s + 1)$.*

*Proof.* See Appendix A.5.1. $\qquad\square$

Therefore, from equation (30) and Lemma A.3, we find that whenever $T > \left(\frac{2}{\eta_3 - \eta_1}\right)^{1/\epsilon} =: T^*(\epsilon, \theta)$,

$$
\sum_{t=1}^{T} \mathbb{P}(v_{2,t} \geq \eta_3, \ k_t \geq L, \ s_t' < \lfloor k_t \eta_1 \rfloor)
$$

$$
\leq \sum_{t=1}^{T} \mathbb{P}\left((\eta_3 - \eta_1)\exp\{k_t d(\eta_1, \eta_3)\} \leq 2t, \ k_t \geq L\right)
$$

$$
\leq \sum_{t=1}^{T} \mathbb{P}\left((\eta_3 - \eta_1)\exp\{L d(\eta_1, \eta_3)\} \leq 2t\right)
$$

$$
= \sum_{t=1}^{T} \mathbb{P}\left((\eta_3 - \eta_1)T^{1+\epsilon} \leq 2t\right) = 0 \tag{31}
$$

All that is left is to bound the second term in (29), and to do so we apply the following Lemma whose proof is in Appendix A.5.2

**Lemma A.4.** *There exist positive constants $C = C(\theta_2, \eta_1)$ and $x' > \theta_2$ such that*

$$
\sum_{t=1}^{T} \mathbb{P}\left(s_t' \geq \lfloor k_t \eta_1 \rfloor, \ \pi_t^{\mathrm{OG}} = 2\right) \leq K + \frac{1}{1 - e^{-d(x', \theta_2)}}
$$

Combining Lemma A.4, (31), (29) and (30) shows the claim. $\qquad\square$

### A.5.1 Proof of Lemma A.3

*Proof.* We upper bound the denominator as follows. Given that $s < \lfloor kx \rfloor$, we have $s \leq kx - 1$. Let $B(a, b)$ denote the Beta function, then

$$
\mathbb{E}\left[(V - y)^+\right] = \frac{1}{B(s+1, k-s+1)} \int_y^1 (t-y)t^s(1-t)^{k-s} \, dt
$$

$$
= \frac{1}{B(s+1, k-s+1)} \int_y^1 t^{s+1}(1-t)^{k-s}dt - y\mathbb{P}\left(V \geq y\right)
$$

$$
= \frac{B(s+2, k-s+1)}{B(s+1, j-s+1)} \left(\frac{1}{B(s+2, k-s+1)}\right) \int_y^1 t^{s+1}(1-t)^{k-s}dt - y\mathbb{P}\left(V \geq y\right)
$$

$$
= \frac{s+1}{k+2}F_{k+2,y}^B(s+1) - y\mathbb{P}\left(V \geq y\right) \tag{32}
$$

$$
\leq \frac{s+1}{k+2}F_{k+2,y}^B(s+1) \leq F_{k,y}^B(kx) \leq \exp\left\{-kd(x,y)\right\} \tag{33}
$$

where we use Fact A.1 and the definition of the Beta CDF to establish equation (32). The final bound in (33) is the result of the Chernoff-Hoeffding theorem and Fact A.3. Let $\delta := y - x$, and note that $s < kx \implies s \leq \lfloor (k+1)x \rfloor$ due to $s$ being integer, whence

$$
\mathbb{E}\left[(y - V)^+\right] = \mathbb{E}\left[(y - V)\mathbb{1}\left\{V \leq y\right\} \mid s, k\right]
$$

$$
= \mathbb{E}\left[(y - V)\mathbb{1}\left\{y - \delta \leq V \leq y\right\} \mid s, k\right] + \mathbb{E}\left[(y - V)\mathbb{1}\left\{V < y - \delta\right\} \mid s, k\right]
$$

$$
> \mathbb{E}\left[(y - V)\mathbb{1}\left\{V < y - \delta\right\} \mid s, k\right]
$$

$$
\geq \delta\mathbb{E}\left[\mathbb{1}\left\{V < y - \delta\right\} \mid s, k\right] \tag{34}
$$

$$
= \delta\mathbb{P}\left(V < x \mid s\right)
$$

$$
= \delta\left(1 - F_{k+1,x}^B(s)\right) \tag{35}
$$

$$
\geq \delta/2 \tag{36}
$$

where equation (35) relies on Fact A.1. The bound (36) is justified from Fact A.2 and $s \leq \lfloor (k+1)x \rfloor$. Thus using the inequalities for both the numerator and denominator, we obtain the desired bound. $\quad\square$

### A.5.2 Proof of Lemma A.4

*Proof.* The steps in this proof follow a similar one in [3] but we show them for completeness. We bound the number of times the suboptimal arm's mean is overestimated. Let $\tau_\ell$ be the time step in which the suboptimal arm is sampled for the $\ell^{\text{th}}$ time. Because for any $x$, $\lim_{n\to\infty} \frac{\lfloor nx \rfloor}{nx} = 1$, we can let $N$ be a large enough integer so that if $\ell \geq N$, then $\eta_1 \frac{\lfloor \ell \eta_1 \rfloor}{\ell \eta_1} > x' := (\theta_2 + \eta_1)/2 > \theta_2$. In that case,

$$
\begin{aligned}
\sum_{t=1}^{T} \mathbb{P}\left( s'_t \geq \lfloor k_t \eta_1 \rfloor, \ \pi_t^{\text{OG}} = 2 \right) &\leq \mathbb{E}\left[ \sum_{\ell=1}^{T} \sum_{t=\tau_\ell}^{\tau_{\ell+1}-1} \mathbb{1}\left\{ s'_\ell \geq \lfloor k_\ell \eta_1 \rfloor \right\} \mathbb{1}\left\{ \pi_t^{\text{OG}} = 2 \right\} \right] \\
&= \mathbb{E}\left[ \sum_{\ell=1}^{T} \mathbb{1}\left\{ s'_{\tau_\ell} \geq \lfloor (\ell-1)\eta_1 \rfloor \right\} \sum_{t=\tau_\ell}^{\tau_{\ell+1}-1} \mathbb{1}\left\{ \pi_t^{\text{OG}} = 2 \right\} \right] \\
&= \mathbb{E}\left[ \sum_{\ell=0}^{T-1} \mathbb{1}\left\{ s'_{\tau_{\ell+1}} \geq \lfloor \ell \eta_1 \rfloor \right\} \right] \\
&\leq N + \sum_{\ell=N+1}^{T-1} \mathbb{P}\left( s'_{\tau_{\ell+1}} \geq \ell \eta_1 \frac{\lfloor \ell \eta_1 \rfloor}{\ell \eta_1} \right) \\
&\leq N + \sum_{\ell=N+1}^{T-1} \mathbb{P}\left( s'_{\tau_{\ell+1}} \geq \ell x' \right) \\
&\leq N + \sum_{\ell=1}^{\infty} \exp(-\ell d(x', \theta_2)) \qquad\qquad (37) \\
&= N + \frac{1}{1 - e^{-d(x', \theta_2)}}
\end{aligned}
$$

$\square$

The bound (37) follows from the Chernoff-Hoeffding theorem and that $s'_{\tau_{\ell+1}} \sim \text{Binomial}(k_{\ell+1}, \theta_2) \sim \text{Binomial}(\ell, \theta_2)$.

### A.6  Additional plots

We include some additional plots that compare Bayes UCB and Thompson Sampling in addition to IDS.

Figure 2: Mean regret in the long horizon Gaussian experiment of section 5.

Figure 3: Mean regret in the corresponding Bernoulli experiment.