[Reviews · NeurIPS 2016]

Reviewer 1

Summary

This paper investigates the retirement value formula defining Gittins' indices, and proposes replacing this optimal stopping problem with a computationally simpler 'optimistic' retirement problem. In the optimistic alternative to the problem, the true mean-reward of the arm is perfectly revealed after a single sample. This always yields an upper bound on the Gittins index of an arm, since the decision-maker is presented with more information. Moreover, it should yield an close approximation for problems with large discount factors, where the quality of an arm can be accurately identified within a negligible portion the effective horizon. The paper's main theoretical result builds on an analysis of Bayes UCB to provide a frequentist regret bound matching the lower bound of Lai and Robbins. In experiments, the algorithm appears to offer state-of-the-art performance for a very widely studied problem.

Qualitative Assessment

I greatly enjoyed this paper. I struggled to select between 3 and 4 for some of the criteria, because there is a big gap between top 30% of submissions and top 3%. I am confident that this paper will have longer lasting impact than the average bandit paper accepted to NIPS. But because the MAB with independent arms has been studied so heavily, it's difficult to make an enormous improvement to the state of the art. It's disappointing that the algorithm depends on a tuning parameter alpha. The algorithm seems robust to choices of alpha, but only in problems where the horizon dwarfs the differences in alpha. Little ways of tuning some of the competing algorithms can make a big difference in performance. It should be made clear that the main strength of IDS, Bayes UCB, and Thompson sampling is that they can be naturally extended to problems with more complicated priors and observation structures. Relative to approximate Gittins' indices, these methods are very flexible. A reasonable narrative for this paper is that we know more about the structure of optimal policies for problems with independent arms, and should exploit that. ------ Notes----- These are provided in case they are helpful. You don't need to address them in the rebuttal. - In the simulation experiments, how does the algorithm perform for shorter horizons? How does it perform with tuning parameter alpha=0? -It's worth pointing out that there is a close conceptual connection between optimistic approximation in your paper, and the expected-improvement algorithm, which is a popular approach in the field of 'Bayesian optimization'. The EI algorithm measures the arm that leads to the largest expected improvement over the current largest posterior mean, assuming the quality of the arm were observed perfectly. This is computed using the expression for the mean of the truncated Gaussian distribution, similar to example 3.2. Your paper merges this into the retirement problem in an interesting way. - Even when you can't calculate the expectation on the right hand side of (4), can you solve this equation using a gradient method? You should be able to bring the derivative inside the expectation, in which case perhaps the derivative can be expressed in terms of the CDF of R(y). I'm not sure. -In lines 102-106, [14] is cited twice, when you mean to cite [13]. - In line 122, just following the definition of the retirement problem, make it clear that the supremum is over stopping rules tau, and not fixed times tau. - In line 159, "Let R(y) be a random variable drawn from the prior..." This is important enough that you I'd consider writing this in math. Also, you should consider changing notation; R(y) suggests strongly that you are drawing a reward, and not a mean, and such an algorithm might perform terribly. -In Prop 3.1, you never explicitly noted that the optimal Bayesian regret is O(log(T)^2) due to Lai, - I've never seen the calculation in example 3.1. Can you provide a reference, or a derivation in the appendix?

Confidence in this Review

3-Expert (read the paper in detail, know the area, quite certain of my opinion)


Reviewer 2

Summary

This paper considers the stochastic multi-armed problem and proposes a new algorithm grounded in Bayesian approach. The algorithm is based on novel "optimistic" approximations to the Gittins index and is shown to match the Lai-Robbins asymptotical lower bound (including the constants) for the frequentist regret. It is also shown in computational simulations that it outperforms the best existing algorithms.

Qualitative Assessment

This is an excellent, well-balanced paper with a very interesting simple idea of approximating the Gittins index, which works suprisingly well in both theory and practice. I have only a couple of minor suggestions: - the idea of an optimistic approximation to the Gittins index relies on Bayesian setting, so I would suggest to make the title more informative, e.g. "Optimistic Gittins Indices for the Bayesian Multi-armed Bandit Problem" - there can be some misleading with the word "optimistic", as there are already "optimistic approach" to describe the inflation term of UCB and alike. It would be desirable to clarify the difference in the introduction. - in Abstract (and elsewhere) it is not clear what "over all sufficiently large horizons" means. Could you give a mathematical definition of this or reformulate it? Are you assuming the discount factor < 1? - it is not clear how the "Bayesian regret" differs from "Bayes-optimality" - line 23: I would say that "index scheme" is not appropriately used in the paper. It would be good to keep the term exclusively for policies which play the arm with highest current value of an index. I.e., randomized policies (TS, UCB variants) are not index rules/schemes/policies/strategies. - "Gittin's" (several times) -> "Gittins" - line 39: "it is unknown whether a Gittins-like index strategy is optimal for a fixed, finite-horizon": Actually, it is known that an index strategy is not optimal for finite horizon (in general). It is easy to obtain the optimal policy using dynamic programming. Inspecting the structure quickly reveals that it is not an index policy. - line 102: citing [14] is probably incorrect. - line 113: Note that while Gittins index is suboptimal, its generalization proposed by Whittle (and described in [16]) is virtually optimal. See also Villar et al 2015 (Statistical Science) - line 118: using q_{i,s} as state is not correct, because it doesn't uniquelly correspond to y_{i,s} - Please add a mathematical proof of Lemma 3.1. It is not bovious why there is a "positive probability that the superior arm is never explored sufficiently..." (line 134) - line 153: "is equivalent" suggest a two-way equivalence (i.e. that also every optimal stopping problem can be solved by computing a Gittins index). This is true (established in Nino-Mora 2007), but not clear from (2). - line 182: is there any difference between \phi and \Phi used in the formula? Please specify what these are - line 185: "and makes" -> "which makes" - line 214: "bounds holds" -> "bound holds" - line 242: specify what possible values the tuning parameter \alpha can take (e.g. >0) - is the Regret reported in Tabels 1 and 2 Bayesian or frequentist? Could you add SD in Table 2? Is there any reason why not to add for comparison also the best non-Bayesian policy? - Figures 1,2,3: these are unreadable when printed in b&w. Please add arrows to indicate which curve corresponds to which policy. Also, instead of just different colors, use different line types. Also use the same format for the same policies in all figures. - many references should have capitalized words (thompson, kl-ucb, ramanuja, ucb, gittins). Also [21] is single-authored.

Confidence in this Review

3-Expert (read the paper in detail, know the area, quite certain of my opinion)


Reviewer 3

Summary

The authors propose a new anytime bandit algorithm based on an efficient approximation of the discounted Gittins index. There are two main ideas. The first is to increase the discount rate as a function of time, which saves the strategy from linear regret. The second is to approximate the Gittins index via a truncation after which the signal noise drops to zero. The primary technical contribution is a theoretical analysis showing the frequentist asymptotic optimality of the strategy when the lookahead is just 1.

Qualitative Assessment

*************************************************** POST REBUTTAL. Thanks for the correction. I agree this makes things OK again. Nice paper. *************************************************** Overall the paper is reasonably well written. I get the feeling the authors removed a lot of intuition to make the paper fit. I'm sympathetic to this plight, but still wish there was a little more there. Even simple things like noting that the effective horizon for discount gamma = 1 - 1/t is O(t) would be helpful. Also, in Examples 3.1/3.2 it would be nice to see some approximate solutions that reveal the strategy has a form approximately the same as some existing strategies (see minor comments). Technically the paper is generally OK, but I do have some possibly serious issues described below. The authors should focus on these in their response to this review to increase my score. * It is claimed at the start of Section 4 that because the strategy is asymptotically optimal in the Lai & Robbins sense that it is also asymptotically Bayesian optimal. Please justify this claim. As far as I know the UCB strategy is not asymptotically Bayesian optimal, but it is optimal in the Lai & Robbins sense. The first index-strategy that is Bayesian optimal is by Lai in '87 (you cite it). I do believe your strategy gets both, but I don't understand the argument. * In L397 in the supplementary material. I don't understand the > in the display. What if q = 1/10 and p = 1/2. Then the first logarithmic term is negative. I tried a little to fix this result and could not. This lemma seems quite critical and I wonder if this is where the improvement is coming from that allows some log factors to be shaved in the confidence interval essentially (I am talking about the discussion in L223). Besides the theoretical results there are also some experiments. I found these to be a bit of a tease because (a) the performance is excellent and (b) we only get to see examples from a two (random) experiments. One Gaussian and one Bernoulli. It's nice to see the quantiles of the distribution, but I would have preferred to see the expected regret over a larger range of problems. Eg., the worst-case regime. Instead the focus is on a very easy problem with few arms and a distribution for which most of the arms are well separated. That said, the results are impressive. Overall I find the paper very interesting. The technical issues force me to lower my score and I wish the execution were better (crisp intuition/experiments). See below for some minor comments. Minor comments: * It might be worth discussing the approximations given in Examples 3.1 and 3.2 in a little more depth. I did not investigate the former, but the latter turns out to be quite interesting. Solving the inequality and doing some naive approximation suggests that the optimal lambda has approximately the form lambda = sqrt(2sigma^2 log(t sigma^2) (here I have chosen gamma = 1 - 1/t as recommended). If the prior is the flat improper Gaussian prior, then in round t we have sigma^2 = 1/N_i(t-1) and one recovers a well-known index, which at least for the Gaussian case has now extremely tight, but not quite optimal (in finite time), regret guarantees. There are also asymptotic guarantees, see [1]. Note that this is basically the algorithm that Lai showed was Bayesian optimal (replace t by n). * L39. I think Berry & Fristedt showed already that the Bayesian strategy is index-based only if geometric discounting and infinite horizon are assumed. * L76. This reference is wrong, although that paper does study the case where the horizon is not known, so may also be relevant. I guess you mean [2] * L216. Not everything that decays faster than 1/t is summable (eg., 1/(t log(t)) * The regret analysis included in the main body is all totally standard, while the interesting stuff is deferred to the appendix. An odd choice? * L208: s_t, s_t' are introduced. Are they ever used? * A bit petty, but I find some of the notation a bit displeasing. For example, T is used for the non-random horizon while k_i(t) is used for the random number of times arm i has been chosen. * Is it written which priors were used by Thompson sampling/Bayes UCB in the experiments? * For what prior is Theorem 1 proven? (looks like flat beta prior, but is it written?). Can you get asymptotic optimality for any prior? Also see [1] for a more general results on BayesUCB References: [1] On Bayesian index policies for sequential resource allocation. Kaufmann, 2015 [2] Regret analysis of the finite-horizon Gittins index strategy for multi-armed bandits. Lattimore, 2016

Confidence in this Review

3-Expert (read the paper in detail, know the area, quite certain of my opinion)


Reviewer 4

Summary

This paper considers the multi-armed bandit problem over all large time horizons, rather than an infinite time horizon. In the latter setting, a famous result is the Gittins index, which is an efficient, optimal solution. However, in this paper's setting, the Gittins index is not proved to work and is computationally inefficient. The authors propose an approximation to the Gittins index that assumes an increasing discount factor. They prove that it matches a known lower bound on regret. Additionally, they empirically demonstrate its advantage over competitors.

Qualitative Assessment

The core contribution of this paper is a notable improvement over past results. The "lookahead" nature of Optimistic Gittins is interesting and innovative. However, there are doubts as to whether the increasing discount factor is appropriate. If I understand correctly, an important assumption, constant discount factor, has been changed, and the paper could be greatly improved with discussion and justification. Since the problem is now different, it is hard to see why it is fair to compare to past results without some argument in the paper. If the increasing discount factor is justified, then this paper could be viewed as quite valuable. There were many small issues with the presentation of this paper. Additionally, the paper could benefit from including results for general K and the exponential family of distributions. L21: Around here might be a good place to discuss some real-world applications of MAB. L36: What is the significance of optimizing over an infinite horizon vs all large time horizons? L92: These two equations are confusing because it is not stated which variables the expectations are taken over. Also, is this "reward" or "expected reward"? L99: Is the lim inf here the reason this problem is considering all large time horizons instead of an infinite horizon? It could benefit the reader to formally state the difference between the two settings. L102: "Another" paper is cited, but the citation number is still [14], the same as the previous paper? L137: Assuming an increasing discount factor could be a significant change to the problem. Could it be discussed and justified more thoroughly? Also, are the results dependent on the rate being asymptotic to 1/t? The assumption seems strong. L139: If the discount factor changes at each step, it seems odd to use Equation 2, which appears to use a constant discount factor. Or is Equation 2 being modified for this strategy? L150: It would help to move the sentences at the very start of 3.1 to here, since "computational burden" is introduced here but explained there. L152: Feels strange to only have one subsection. Perhaps call everything before 3.1 and change this to 3.2 L195: "In the sequel" is an outdated phrase. A better alternative would be "in the following." L201: Has \theta^* been defined yet? L202: This line is redundant. L211: This equation needs an explanation. At least the interpretation of L(T) should be described. Also, in the last line, how does d(\theta_2,\theta_1) gain the (\theta_1-\theta_2) factor in (5)? L241: This note would be better placed at the introduction of the increasing discount factor. L255: It may be informative to mention the specifications of the CPU. Table 1 and 2: Please boldface the best result in each row. Also, "Algorithm" should be something like "Metric" because its row is for column headers, but "Algorithm" is a row header. L283: If this section is to be included, it seems necessary to make space for the graphs from A.6. Otherwise the paragraph feels like it has little impact. All these experiments should show hypothesis tests or confidence intervals. The lack of conclusion is jarring.

Confidence in this Review

2-Confident (read it all; understood it all reasonably well)


Reviewer 5

Summary

This paper proposed approximations to Gittins index with lower computational burden, with a competitive performance on regret lower bound.

Qualitative Assessment

1. Please refine the paper. For example, in lines 96 and 102, reference 14 was mentioned, but one of them should be the other paper. 2. An table to show the pros and cons of the mentioned index schemes in different situations (MABP on iid or Markov RV, for example) should be added. 3. For figure 1, error bars should be added.

Confidence in this Review

2-Confident (read it all; understood it all reasonably well)